# A Retrospective Analysis of 2-Year Follow-Up of Patients with Incidental Findings of Sarcoidosis

**DOI:** 10.3390/diagnostics14030237

**Published:** 2024-01-23

**Authors:** Oluwabukola Thomas-Orogan, Shaney L. Barratt, Muhammad Zafran, Apollo Kwok, Anneliese Simons, Eoin P. Judge, Matthew Wells, Richard Daly, Charles Sharp, Abiramy Jeyabalan, Martin Plummeridge, Ladli Chandratreya, Lisa G. Spencer, Andrew R. L. Medford, Huzaifa I. Adamali

**Affiliations:** 1Bristol Interstitial Lung Disease Service, North Bristol NHS Trust, Bristol BS10 5NB, UK; kola.thomas@nhs.net (O.T.-O.); shaney.barratt@nbt.nhs.uk (S.L.B.); martin.plummeridge@nbt.nhs.uk (M.P.); 2Liverpool Interstitial Lung Disease Service, Liverpool University Hospitals NHS Foundation Trust, Liverpool L7 8XP, UK; apollo.kwok@liverpoolft.nhs.uk (A.K.); anneliese.simons@yahoo.co.uk (A.S.);; 3Department of Cellular Pathology, North Bristol NHS Trust, Bristol BS10 5NB, UK; richard.daly@nbt.nhs.uk; 4Department of Radiology, North Bristol NHS Trust, Bristol BS10 5NB, UK; ladli.chandratreya@nbt.nhs.uk; 5North Bristol Lung Centre, North Bristol NHS Trust, Bristol BS10 5NB, UK; andrew.medford@nbt.nhs.uk

**Keywords:** sarcoidosis, thoracic lymphadenopathy, pulmonary sarcoidosis, follow-up

## Abstract

Introduction: Sarcoidosis is a multi-system granulomatous disease most commonly involving the lungs. It may be incidentally diagnosed during imaging studies for other conditions or non-specific symptoms. The appropriate follow-up of incidentally diagnosed asymptomatic stage 1 disease has not been well defined. Objective: To define the clinical course of incidentally diagnosed asymptomatic stage 1 sarcoidosis and propose an algorithm for the follow-up of these patients. Methodology: A retrospective case note analysis was performed of all EBUS-TBNA (endobronchial ultrasound-guided transbronchial needle aspiration)-confirmed cases of stage 1 sarcoidosis presenting incidentally to Bristol and Liverpool Interstitial Lung Disease services. Clinical history, serology results, imaging scans, and lung function parameters were examined at baseline, 12, and 24 months. A cost analysis was performed comparing the cost of the current 2-year follow-up guidance to a 1 year follow-up period. Results: Sixty-seven patients were identified as the final cohort. There was no significant change in the pulmonary function tests over the two-year follow-up period. Radiological disease stability was observed in the majority of patients (58%, *n* = 29), and disease regression was evidenced in 40% (*n* = 20) at 1 year. Where imaging was performed at 2 years, the majority (69.8%, *n* = 37) had radiological evidence of disease regression, and 30.2% (*n* = 16) showed radiological evidence of stability. All patients remained asymptomatic and did not require therapeutic intervention over the study period. Conclusions: Our results show that asymptomatic patients with incidental findings of thoracic lymph nodal non-caseating granulomas do not progress over a 2-year period. Our results suggest that the prolonged secondary-care follow-up of such patients may not be necessary. We propose that these patients are followed up for 1 year with a further year of patient-initiated follow-up (PIFU) prior to discharge.

## 1. Introduction

Sarcoidosis is a multi-system, non-caseating, granulomatous disease of unknown aetiology that most commonly presents with pulmonary involvement [1]. Isolated thoracic lymphadenopathy is usually asymptomatic and may be incidentally identified during screening or following investigations for other conditions and/or non-specific symptoms. It has been estimated that between 30 and 60% [1,2] of patients with sarcoidosis present without symptoms, although the true prevalence of asymptomatic disease is difficult to determine due to population differences in environmental exposures, surveillance methods, ethnicity, and predisposing genetic factors.

The treatment of sarcoidosis is usually indicated where there is either a high risk of mortality and/or morbidity due to major organ involvement or an unacceptable loss of quality of life. As such, isolated thoracic lymphadenopathy may not warrant therapy. The duration and modalities for follow-up in these patients has not been standardised [3], although the recent British Thoracic Society (BTS) clinical statement on pulmonary sarcoidosis has recommended that patients with stage 1 disease based on a chest X-ray are monitored every 6 months for two years [4].

This study aimed to define the clinical course of asymptomatic, thoracic lymph nodal sarcoidosis. Specifically, we aimed to determine whether patients develop features of progressive disease and whether protracted secondary-care follow-up is required. We then performed a cost analysis for the introduction of a shorter follow-up duration for these patients. Finally we aimed to develop an algorithm for the follow-up of these patients.

## 2. Materials and Methods

### 2.1. Study Design

This was a retrospective, multi-centre, observational cohort study undertaken at two large secondary-care institutions in the UK providing specialist interstitial lung disease services (Bristol Interstitial Lung Disease Service, North Bristol NHS Trust and Liverpool Interstitial Lung Disease Service, Liverpool Hospitals NHS Foundation Trust). The study was approved by the Health Research Authority, United Kingdom (IRAS 274055).

### 2.2. Study Subjects

Consecutive patients referred to the above centres between November 2010 and February 2020 with incidentally identified stage 1 sarcoidosis were included. All cases demonstrated non-caseating granulomas on histology obtained from EBUS-TBNA (endobronchial ultrasound-guided transbronchial needle aspiration), with the exclusion of tuberculosis through specimen culture and immunohistochemical staining. EBUS-TBNA was performed at our tertiary EBUS centres as previously described [5]; the bronchoscopy did not reveal evidence of endobronchial sarcoid in all of these patients. All diagnoses of sarcoidosis were confirmed via multidisciplinary team consensus and in accordance with the joint statement of the American Thoracic Society, the European Respiratory Society, and the World Association of Sarcoidosis and Other Granulomatous Disorders (ATS/ERS/WASOG) [6]. The patients were labelled as asymptomatic after the investigations (EBUS-TBNA and imaging) had been performed; the lung parenchyma in all these patients were normal, and pulmonary sarcoidosis was therefore excluded as a cause for any symptoms, which were then attributed to other known co-morbidities. The patients had no features of extra-pulmonary sarcoidosis.

### 2.3. Outcome Measures

Stage 1 sarcoidosis was defined based on the presence of thoracic lymphadenopathy identified via chest X-ray and the absence of parenchymal disease, as determined by a consultant thoracic radiologist. Some patients also underwent high-resolution computed tomography (HRCT) scans, which were used as confirmatory scans.

The primary outcome was evidence of progression, as defined based on an increase in radiological Scadding stage, as determined via chest X-ray. Regression was defined by the resolution of lymphadenopathy, as determined via chest X-ray.

Data were collated on baseline patient demographics (age, gender, comorbidities). Serology results, pulmonary function testing, and radiological stage at diagnosis, 1 year (+/− 3 months), and 2 years (+/− 3 months) from diagnosis were also collated.

### 2.4. Cost Analysis

Figures from the 2022/23 National Tariff Payment System were used as a surrogate for cost. We compared the average cost of following our patients up for two years, as is currently recommended, against a model where these patients are discharged after 1 year. We considered costs for HRCT, clinic appointment with a respiratory physician, full pulmonary function testing, and respiratory physiology appointment. We excluded costs for chest X-rays and blood tests, which were negligible. We also excluded costs for EBUS-TBNA as we assumed that all patients in both comparison arms would have one each, hence cancelling out the cost.

Payment by Results (PbR) is the payment system in England enabling healthcare commissioners to reimburse healthcare providers for each patient seen or treated. National tariffs are set annually based on the average cost of services reported by NHS providers, considering the complexity of the patient’s needs. The PbR tariffs (2022/23) were used to calculate costs to the healthcare funding entity, in this case, the Clinical Commissioning Group (CCG). Health Resource Group (HRG) codes are used within the NHS to assist calculation of reimbursement for procedures. The HRG codes used in this study were RD20A (Computerised Tomography Scan of One Area, Without Contrast, 19 Years and Over) and DZ52Z (Full Pulmonary Function Testing).

### 2.5. Statistical Analysis

Categorical variables were presented as counts and percentages. All continuous variables were non-parametric and therefore were presented as medians and interquartile range (IQR). Differences between patient groups were evaluated using the Friedman test. For all tests, a *p* ≤ 0.05 was considered statistically significant. Data were analysed using IBM SPSS Statistics for Windows, version 28 (IBM Corp., Armonk, NY, USA).

## 3. Results

### 3.1. Patient Demographics

A total of 339 patients with EBUS-TBNA-confirmed sarcoidosis were identified (see Figure 1). Two hundred and seventy-two of these were excluded for the following reasons: additional parenchymal involvement (*n* = 92), symptomatic disease and/or extrapulmonary involvement (*n* = 53), EBUS-TBNA performed as a tertiary referral out of area with missing data (*n* = 83), therapeutic intervention (*n* = 22), and the presence of concurrent malignancies and tuberculosis (*n* = 22).

The final cohort consisted of 67 patients who were asymptomatic and were incidentally identified as having stage 1 thoracic sarcoidosis. Table 1 summarises the baseline characteristics. The majority of the cohort were British Caucasian (82.1%, *n* = 55), non-smokers (64.2%, *n* = 43), and male 52.2%, *n* = 25), with a median age of 54 years (IQR 47–63.5). Serum angiotensin converting enzyme (ACE) levels were elevated in over half of the cohort (51.9% *n* = 28), and serum calcium levels were normal in all patients.

One-year follow-up pulmonary function tests and chest X-rays were performed at a median of 12.5 (8.8–16) months and 12.4 (8.3–15) months, respectively, whilst 2-year follow-up pulmonary function tests and chest X-rays were performed at a median of 23 (20.5–28) and 25 (24–35.5) months, respectively.

### 3.2. Co-Morbidities, Symptoms, and Serology

Patients were predominantly asymptomatic; however, some patients presented with non-specific symptoms, including cough (*n* = 11), shortness of breath (*n* = 10), chest pain (*n* = 2), and fatigue (*n* = 2). These were attributed to their known co-morbidities after investigations confirmed sarcoidosis with no parenchymal or endobronchial involvement. Hypertension and asthma were the most common co-morbidities observed in our cohort (18.2%, *n* = 10 each). Other co-morbidities included obstructive sleep apnoea (9%, *n* = 6), type 2 diabetes mellitus (9%, *n* = 6), obesity (4.5%, *n* = 3), and chronic obstructive pulmonary disease (1.8%, *n* = 1).

Fifty-two percent (*n* = 28) of the cohort had elevated ACE levels at presentation. Some patients (*n* = 12) had repeat ACE at one year and, of these, 33% (*n* = 4) had elevated levels. Serum calcium levels remained normal in all patients throughout the study period.

Renal and liver function tests showed no significant change over the follow-up period.

### 3.3. Radiological Changes

The radiological changes observed over the 2-year follow-up period are illustrated in Table 2. Chest X-ray imaging modality was used, with some patients having confirmatory HRCT scans, as seen in Table 3. At the 1 year follow-up, most patients had radiological evidence of disease stability (58%, *n* = 29), whilst 40% (*n* = 20) of patients demonstrated disease regression. Where imaging was available at 2 years, the majority (69.8%, *n* = 37) had radiological evidence of disease regression, and 30.2% (*n* = 16) showed radiological evidence of stability.

In one patient, there was evidence of radiological disease progression to stage 2 at one year, but the patient remained asymptomatic, with no impairment in pulmonary function or evidence of extrapulmonary disease and, therefore, this patient was not commenced on treatment. The two-year follow-up chest X-ray and HRCT showed the complete resolution of both the lymphadenopathy and parenchymal infiltration. In addition, the corresponding two-year lung function tests and serological tests remained normal, and the patient remained asymptomatic.

### 3.4. Pulmonary Function Tests

Most patients had normal pulmonary function at baseline. Significant respiratory co-morbidities, such as pre-existing asthma and obstructive sleep apnoea, were reported in those patients presenting with abnormal pulmonary function.

There was no statistically significant difference in pulmonary function tests over the 2-year follow-up period. Table 4 summarises the changes seen over the two-year period.

### 3.5. Cost Analysis

The cost of following up our cohort over 2 years was an average of GBP 1245 per patient. This included the costs for an average of 3.9 clinic appointments, 1.5 HRCTs, and 1.7 pulmonary function tests with respiratory physiologists. The cost saving from discharge after 1 year would be GBP 440 per patient, based on an average of two clinic appointments, one pulmonary function test, and one HRCT. This would represent a total saving of GBP 29480 to the CCG based on our cohort of 67 patients.

### 3.6. Algorithm

Taking into consideration the above results and costs, we propose a follow-up algorithm for managing incidental histologically confirmed stage 1 sarcoidosis patients who remain completely asymptomatic (Figure 2). The current monitoring model requires frequent follow-up and testing, which, based on our British cohort, may not be necessary. As a result, we propose that these patients are discharged through a patient-initiated follow-up (PIFU) scheme, which will allow re-referral if there are concerns or symptoms suggestive of active sarcoidosis. Patient education plays an important role, and other caveats to this algorithm include the consideration of ethnicity and engagement; Afro-Caribbean patients may require the more traditional follow-up pathway.

## 4. Discussion

This study examined the clinical, radiological, and serological features of patients with incidentally diagnosed stage 1 sarcoidosis over a 2-year period in two large secondary-care institutions in the UK. There was no clinically significant disease progression in this cohort, and all patients remained asymptomatic, without requiring therapeutic intervention, throughout the study period.

Incidental thoracic lymphadenopathy often presents in asymptomatic patients or as part of non-specific symptoms. It presents interesting challenges for respiratory physicians to decide on further diagnostic steps. These incidental lymph nodes are usually detected when screening for coronary artery disease [7,8,9] and lung cancer [10,11]. Stage 1 sarcoidosis was confirmed in 50% of Scandinavian patients discovered as part of general health screening [12] and was biopsy-confirmed in 22% of patients in one study of patients with incidental mediastinal lymphadenopathy found via chest imaging [13]. Similarly, our cohort had node-limited disease and had incidental findings of EBUS-TBNA-proven sarcoidosis; they did not display sarcoid symptoms and manifestations of extra-thoracic disease. Our patients were slightly older than those described in other studies [12,14,15] and registries [16,17,18]. There was a lower proportion of males, but variability in gender distribution has been noted across studies. Like in other studies, the majority of patients in our cohort had never smoked (64.2%) [16,19].

Sarcoidosis is a benign disease, and in the majority of patients, there is the regression or stabilisation of pulmonary disease. However, there is always the concern of progression of disease as described in other French [20], Swedish [21], and UK [15] sarcoid cohorts. Reassuringly, between 55 and 90% of patients with stage 1 disease show spontaneous regression within the first two years of diagnosis [4,6]. Our results showed radiological disease regression in 40% of patients at 1 year and 69.8% of patients at 2 years. Only one patient showed radiological evidence of disease progression to stage 2 but remained asymptomatic with no serological abnormalities and/or impaired pulmonary function tests. This is consistent with the findings showing discordance between radiological stage and pulmonary function trends [22]. The absence of accompanying clinical symptoms or worsening serological and physiological parameters provides certainty that this patient did not progress clinically.

The majority of patients with sarcoidosis have normal lung function but may also show restrictive [23,24] or obstructive patterns [1,25]. Recent evidence suggests that the dominant functional abnormality in intrathoracic sarcoidosis is obstruction, affecting the entire length of the bronchial tree and causing a wide range of airways impairment and altered gas diffusion [26]; the obstructive disturbances are present from the early stages, mainly affecting small airways, and, in advanced disease, obstructive impairment often exists with restrictive disorders. The evidence of reduction in DLCO may suggest parenchymal involvement or indicate the presence of pulmonary hypertension [27,28]. Although the changes in FVC over time only partially correlate with HRCT findings, a combined analysis with DLCO is likely to be more accurate for assessing the progression of the disease [29]. In our cohort, we showed that lung function was normal for the majority of patients unless they had a pre-existing airway disease. Over the two-year period, there was no change in lung function, suggesting that serial monitoring may not be warranted in stage 1 disease unless there is suspicion of the progression of the disease as suggested based on serial imaging.

The use of chest X-rays and the radiographic staging system for the initial evaluation of patients has been a topic of debate, six decades after its introduction by Wurm [30] and modification by Scadding [31]; the poor sensitivity and impaired inter-observer agreement of chest X-rays serve as significant disadvantages to its use [32]. HRCT scans were seen to show pulmonary involvement in 23% of patients with stage 0 and 30% of patients with stage 1 on chest radiographs in a recent study [33]. Furthermore, HRCT provides the ability to identify active inflammation (nodules, ground-glass opacities, alveolar opacities, interlobular septal thickening, and intralobular opacities) from established fibrosis (bullae, broad and coarse septal bands, architectural distortion, volume loss, traction bronchiectasis, and honeycombing) [34]. The recent British Thoracic Society clinical statement recommends that chest X-rays are used for initial evaluation, whilst HRCT scans are limited to patients with an unexplained deterioration in clinical features and pulmonary physiology, allowing the assessment of lung parenchymal changes [4]. Most patients in our cohort had chest X-rays throughout the follow-up period, with a few having corresponding HRCT scans (Table 3). There was general concordance between the chest X-ray and HRCT results, with no additional sarcoid features found on the HRCT scans. Although the need for serial HRCT scans in sarcoidosis monitoring is questioned due to the exposure to high radiation levels, the higher sensitivity provides a significant advantage. HRCT scoring methods have been explored in research but are yet to be adopted clinically [35].

Our cohort showed elevated ACE in 51.9% of patients who were deemed asymptomatic. The unreliability of serum ACE levels in monitoring sarcoidosis has been reported previously even in self-resolving cases [36]; it has been found not to correlate with symptoms, pulmonary function test results or radiological findings and, therefore, should not be used to diagnose or determine prognosis but are used more as a supportive test [36]. In self-resolving disease, the serum ACE may remain elevated and should not cause concern. We also examined the presence of hypercalcaemia since it is evident in patients with active disease; none of our patients showed evidence of this.

A shorter follow-up period also provides significant cost savings to the CCGs. Our cost analysis using data from two ILD centres suggests an average saving of GBP 440 per patient if discharged after 1 year following two clinic appointments, one pulmonary function test, and one HRCT. It is important to mention that our results most likely underestimate the potential savings, as the service across both centres already reflects a change to fewer follow-up visits for these patients. To provide reassurance to clinicians, a more cautious approach could be placing these patients on patient-initiated follow-up (PIFU) for a further 12 months following their 1 year follow-up visit (Figure 2). This allows these patients to take control of their condition and get back in touch if they deteriorate whilst avoiding unnecessary follow-up visits and missed appointments with their associated costs.

We acknowledge the limitations of this study being a retrospective observational study. These include missing data and a small patient cohort. Specifically, 2-year follow-up chest X-rays are missing for 14 patients in our cohort. These patients were followed up clinically but were not deemed to have any suggestion of active disease at this point; therefore, imaging was not performed. Differing practices across sites and the retrospective nature of the study meant that baseline and follow-up investigations, such as ECG, vitamin D25 and 1.25, and urinary calcium were not ubiquitously performed in all patients. These limitations can be addressed by developing new databases and improving the current ones to ease the longitudinal follow-up of patients in future studies. This study was also largely dependent on the limits of the sensitivity offered by chest X-rays, which is known to be poor. A well-developed HRCT scoring system similar to the Scadding criteria might provide a more sensitive prognostic tool.

Furthermore, it is important to note the possible confounding effects of our cohort’s ethnicity on the results. It is possible that the milder outcomes observed in this study are, in part, due to a mainly British Caucasian population cohort (82.1%). Studies have shown a better prognosis of disease in white patients compared to other ethnicities [37].

There are also limitations associated with the cost analysis. We have used tariffs from the 2022/23 National Tariff Payment System as a surrogate for costs. This reflects the tariffs for this time point, which will have changed over the past 10 years and will in due course be replaced. Furthermore, this cost analysis is only directly applicable to the UK health system; however, the same principles underlying the calculations could be applied to other healthcare funding systems.

In summary, this study shows no clinically significant disease progression in patients with incidentally diagnosed asymptomatic stage 1 EBUS-TBNA-proven sarcoidosis. Our results suggest that the continued secondary-care follow-up of these patients may not be necessary and, rather, PIFU may have a role to play. This would alleviate the burden on patients, reduce care provision costs, and reduce the pressures on healthcare services, especially in the age of COVID-19. This study may provide guidance for future consultations; however, further work with a larger and more diverse cohort is required to explore the possible reasons underpinning this finding. This may require collaboration from pan-global sarcoid registries.

## Figures and Tables

**Figure 1 diagnostics-14-00237-f001:**
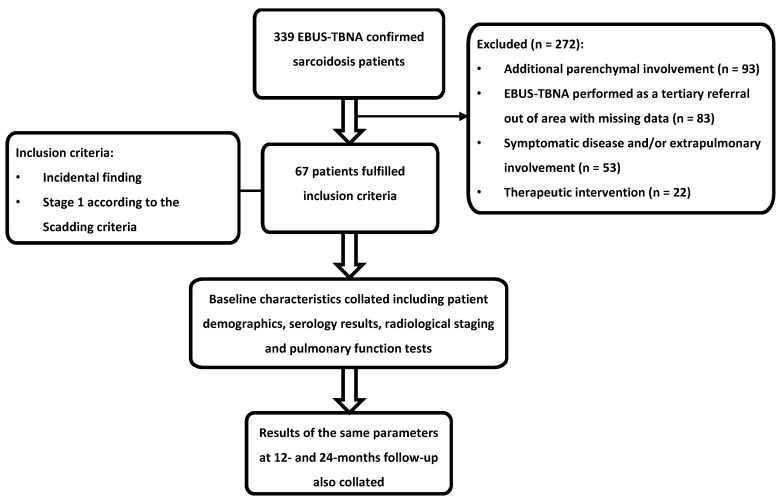
Study methodology. Two hundred and seventy-two patients were excluded for not meeting the inclusion criteria. EBUS-TBNA, endobronchial ultrasound-guided transbronchial needle aspiration.

**Figure 2 diagnostics-14-00237-f002:**
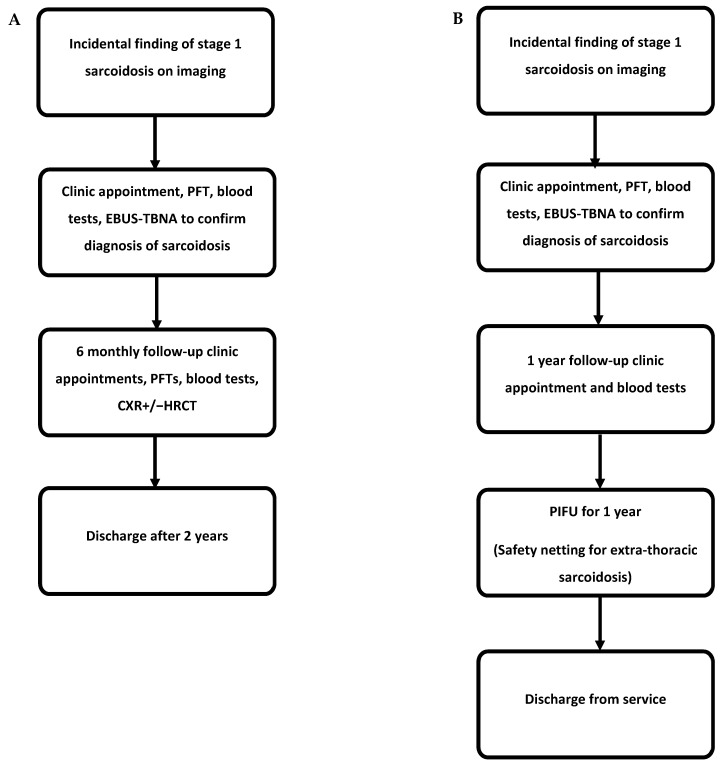
Follow-up algorithms. (**A**) Algorithm based on the current follow-up guidance. (**B**) Suggested algorithm for follow-up incorporating PIFU. Both algorithms assume no progression of the disease over the follow-up period. PFT, pulmonary function test; HRCT, high-resolution computed tomography; EBUS-TBNA, endobronchial ultrasound-guided transbronchial needle aspiration; CXR, chest X-ray; PIFU, patient-initiated follow-up.

**Table 1 diagnostics-14-00237-t001:** Baseline characteristics of patients. N, number; IQR, interquartile range; ACE, angiotensin-converting enzyme; l, litres; FEV1, forced expiratory volume in 1 s; FVC, forced vital capacity; TLCO, transfer factor of the lung for carbon monoxide; mmol, millimoles; kPa, kilopascal; KCO, carbon monoxide transfer coefficient. Data presented as median (IQR) unless otherwise stated.

Characteristic		N
**Male, % (n)**	52.2% (35)	67
**Age in years (IQR)**	54 (47–63.5)	67
**Smoking status, % (n)**		67
Non-smoker	64.2 (43)	
Ex-smoker	17.9 (12)	
Current smoker	7.5 (5)	
Unknown	10.4 (7)	
**Ethnicity, % (n)**		67
British Caucasian	82.1 (55)	
British Black	1.5 (1)	
British Asian	3.0 (2)	
Egyptian	1.5 (1)	
Unknown	11.9 (8)	
**Serum calcium**		56
Normal, % (n)	100 (56)	
**Serum ACE**		54
Normal, % (n)	48.1 (26)	
Elevated, % (n)	51.9 (28)	
**eGFR (ml/min)**	78.5 (65.5–88.8)	61
**Liver function tests**		62
Albumin	39 (37–43)	
Bilirubin (µmol/L)	7 (6–9)	
ALT (U/L)	23 (20–36)	
ALP (U/L)	77 (61.3–97.5)	
**FEV1, l**	2.9 (2.2–3.5)	56
**FEV1, % predicted**	98.4 (88.6–112.6)	55
**FVC, l**	3.7 (3.0–4.7)	55
**FVC, % predicted**	107 (99.3–124.8)	54
**FEV1/FVC, %**	76 (72–80)	54
**TLCO, mmol/min/kPa**	7.9 (6.4–9.5)	38
**TLCO, % predicted**	90 (76.7–102.8)	38
**KCO, mmol/min/kPa/l**	1.5 (1.4–1.6)	37
**KCO, % predicted**	93 (87–106)	37

**Table 2 diagnostics-14-00237-t002:** Chest X-ray changes over the 2-year follow-up period. Regression was defined based on the resolution of lymphadenopathy. N, number. Data presented as percentages and counts.

	1-Year Follow-Up Scan	n	2-Year Follow-Up Scan	n
**Proportion of patients that remained stable, % (n)**	58 (29)	50	30.2 (16)	53
**Proportion of patients that regressed, % (n)**	40 (20)		69.8 (37)	
**Proportion of patients that progressed, % (n)**	2 (1)		0 (0)	

**Table 3 diagnostics-14-00237-t003:** Imaging follow-up over 2-year period. N, number. Data presented as counts.

	Baseline	1-Year Follow-Up	2-Year Follow-Up
**Chest X-rays completed (n)**	67	50	53
**HRCTs completed (n)**	67	19	13

**Table 4 diagnostics-14-00237-t004:** Pulmonary function test changes over the 2-year follow-up period. Statistical analysis was performed using the Friedman test. A *p* value ≤ 0.05 was considered statistically significant. FEV1, forced expiratory volume in 1 s; FVC, forced vital capacity; TLCO, transfer factor of the lung for carbon monoxide; KCO, carbon monoxide transfer coefficient. Data presented as median (IQR).

	Baseline	1-Year Follow-Up	2-Year Follow-Up	
				*p* Value
**FEV1, % predicted**	98.4 (88.6–112.6)	96.9 (86.2–113)	99 (91.8–116.8)	0.437
**FVC, % predicted**	107 (99.3–124.8)	105 (94.5–126)	110 (99.3–131.3)	0.699
**TLCO, % predicted**	90 (76.7–102.8)	87.6 (77.3–103)	86 (70.6–95)	0.124
**KCO, % predicted**	93 (87–106)	96.5 (86.3–105.8)	92.2 (81.2–106)	0.195

## Data Availability

The data that support the findings of this study are available upon request from the corresponding author, H.I.A.

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
