# Peer review of "A Retrospective Analysis of 2-Year Follow-Up of Patients with Incidental Findings of Sarcoidosis"

_diagnostics, 2024, doi:10.3390/diagnostics14030237_

Round 1

Reviewer 1 Report

Comments and Suggestions for Authors

Thank you for the opportunity to review this interesting and well written original article. Authors in this article assess the 2-year follow-up of patients with incidental findings of sarcoidosis. According to the authors asymptomatic patients with incidental findings of thoracic lymph nodal non-caseating granulomas do not progress over a 2-year period. Based on elementary statistical analysis, it is suggested that prolonged secondary care follow up of such patients may not be necessary. A plain idea that could lead to cost effectiveness.

Author Response

Dear reviewer,

Many thanks for taking the time to review our submitted article and provide your comments.

Kind regards,

Matthew Wells & Huzaifa Adamali

Reviewer 2 Report

Comments and Suggestions for Authors

In their manuscript, Thomas-Orogan et al. explore the evolution of stage 1 sarcoidosis over a two-year period in a cohort of 67 patients followed in two interstitial lung disease clinics. They observe that in the roughly 80% of patients for whom they had completed follow-up, none showed progression of Scadding score and pulmonary manifestations after two years. In the name of cost reduction, they further propose an algorithm by which patients are monitored for one year, and if stable, left to self-monitor, and if stable after two years, “discharged from service” outright.

The retrospective nature of the study carries usual limitations, but notable strengths of the study design are that only patients with pathologically diagnosed disease were included and that the monitoring of the patients appear to have been rigorously uniform. With some revision, I believe that the paper can be accepted for publication.

There are some limitations and critiques that should be addressed.

1.       My biggest concern is the loss of over 20% (14 patients) of their original cohort after two years, a fairly large number. What happened to them? Is it possible that they developed non-pulmonary disease that caused them to have care taken up in other clinics? The authors should address this limitation and comment on how this attrition might affect their conclusions and proposals.

2.       While there was uniformity in the screening and monitoring for pulmonary disease, how thoroughly and consistently was assessment of non-pulmonary disease at the outset and through the course of the study period?  Did all patients uniformly get ophthalmologic examinations, EKGs, vitamin D 25 and 1,25, urinary calcium, etc.? If not, can it be concluded that the patients truly had only intra-thoracic lymph node disease?

3.       This manuscript, like the majority of papers written about sarcoidosis, views unconsciously with a “lung-centric” lens; this is a historical phenomenon. Commenting as a rheumatologist who sees a different segment of the sarcoidosis universe, I am not comfortable with blanket “PIFU” and “Discharge from service”.  Perhaps it might be true that from a pulmonary medicine perspective, patients might be left to their own recognizance, but there is more to sarcoidosis than simply intra-thoracic disease. Many patients who are diagnosed with stage 1 disease and who do not have progression of Scadding score nonetheless regularly develop manifestations in other organ systems sometimes years later and sometimes with severe implications. Readers of this article should not be left with the impression that patients with sarcoidosis can be “discharged” after two years of stable Scadding scores with a clean bill of health and without ongoing follow-up.

4.       More out of curiosity than anything else, it would have been interesting to know more about the baseline characteristics of the patients that were excluded.  For example, is ethnic distribution different in patients with more disseminated disease?

5.       In the Introduction, I believe that the word “prevalence” is more appropriate than “incidence.”

Comments on the Quality of English Language

1.       Some minor style issues may need to be addressed.

a.       The authors sometimes use lower case and other times use upper case lettering for the same words (e.g., Stage, Tuberculosis, Chest X-ray, Angiotensin Converting Enzyme, etc.); they should be consistent.

b.       I have reworded a couple of awkward sentences. From the Introduction: “Treatment of sarcoidosis is usually indicated when there is either high risk of mortality and/or morbidity due to major organ involvement or if there is unacceptable loss of quality of life.” From the Discussion: “Our patients were slightly older than those described in other studies and registries. There was a lower proportion of males, but variability in gender distribution has been previously noted across studies.”

Author Response

Dear reviewer,

Many thanks for taking the time to consider our manuscript for publication, and providing your helpful and constructive comments, which we shall address below:

1. My biggest concern is the loss of over 20% (14 patients) of their original cohort after two years, a fairly large number. What happened to them? Is it possible that they developed non-pulmonary disease that caused them to have care taken up in other clinics? The authors should address this limitation and comment on how this attrition might affect their conclusions and proposals.

A small proportion of these patients were referred from peripheral district general hospitals and as a result, were followed up locally. We have amended the manuscript to highlight the 'hub & spoke' model of referrals to the 2 tertiary centres and now specifically acknowledge the loss of 14 patients chest X-ray imaging at 2 year follow up. We emphasize that these patients were indeed reviewed clinically, however differing practice between centres meant that asymptomatic and stable patients may not have had routine imaging performed, in particular with these few patients.

2. While there was uniformity in the screening and monitoring for pulmonary disease, how thoroughly and consistently was assessment of non-pulmonary disease at the outset and through the course of the study period?  Did all patients uniformly get ophthalmologic examinations, EKGs, vitamin D 25 and 1,25, urinary calcium, etc.? If not, can it be concluded that the patients truly had only intra-thoracic lymph node disease?

We are again limited by the multicentre and retrospective nature of our work. Whilst all patients will have received a thorough multisystem assessment in clinic for symptoms, investigations for baseline and follow up EKG, vitamin D, urinary calcium were not ubiquitously performed, unless relevant symptoms were described. We agree that this is an important consideration for patients who have been labelled asymptomatic and this is now reflected in our manuscript. This lends weight to our suggestion that new or amended sarcoidosis databases may contribute to higher quality research moving forwards. 

3. This manuscript, like the majority of papers written about sarcoidosis, views unconsciously with a “lung-centric” lens; this is a historical phenomenon. Commenting as a rheumatologist who sees a different segment of the sarcoidosis universe, I am not comfortable with blanket “PIFU” and “Discharge from service”.  Perhaps it might be true that from a pulmonary medicine perspective, patients might be left to their own recognizance, but there is more to sarcoidosis than simply intra-thoracic disease. Many patients who are diagnosed with stage 1 disease and who do not have progression of Scadding score nonetheless regularly develop manifestations in other organ systems sometimes years later and sometimes with severe implications. Readers of this article should not be left with the impression that patients with sarcoidosis can be “discharged” after two years of stable Scadding scores with a clean bill of health and without ongoing follow-up.

Thank you for highlighting this important point. We agree that the risk of recrudescence, including within other organ systems, persists in sarcoidosis patients. However, given 70% of recognised cases are confined to thoracic lymphadenopathy (and presumably a higher percentage that never comes to medical attention), we do feel that PIFU is an appropriate use of resources, provided the patient is deemed at 'low risk' and received relevant sarcoidosis specific information and written materials pertaining to extra-thoracic disease. The respiratory clinic is not the optimal environment to follow asymptomatic patients for the possibility they may develop extra-thoracic disease.

We agree with your concern that the manuscript may be interpreted as falsely reassuring, and as such have added a caveat to figure 2 - 'Safety netting for extra-thoracic sarcoidosis'. We further acknowledge this within the new section '3.6 Algorithm' stressing that patient education and engagement needs consideration before issuing a PIFU, which as you suggest may not be appropriate for all patients seen.

4. More out of curiosity than anything else, it would have been interesting to know more about the baseline characteristics of the patients that were excluded.  For example, is ethnic distribution different in patients with more disseminated disease?

We are sorry that we are unable to provide this information; following exclusion data was not collected.

5. In the Introduction, I believe that the word “prevalence” is more appropriate than “incidence.”

Thank you - amended

With respect to the style issues, these have all been addressed.

We hope that in light of the suggested changes we have made, you agree that the manuscript reads better and has a greater acknowledgement of the limitations of the work. 

Kind regards,

Matthew Wells and Huzaifa Adamali

Reviewer 3 Report

Comments and Suggestions for Authors

Manuscript title "A retrospective analysis of 2-year follow-up of patients with incidental findings of sarcoidosis"
1) The authors present a retrospective study defining the clinical course of incidentally detected pulmonary sarcoidosis. A cost analysis was performed, with a new follow-up algorithm being proposed as a result.
2) The study is relevant to pulmonologists and healthcare managers.
3) The main study strength is detailed methodology, allowing for a high confidence in reproducibility. The weaknesses include a perceived mismatch between study title, goal and conclusion with the algorithm appearing rather suddenly. Was the development of a new algorithm not the end goal?
4) The tables and figures are informative.

Author Response

Dear reviewer,

Many thanks for taking the time to review our submitted article and provide your comments. We acknowledge the introduction of the algorithm is somewhat sudden and have modified the abstract to suggest this is the direction of the discussion and manuscript. We have also added section 3.6 - Algorithm which serves as an introduction to the proposed algorithm, further described in the discussion.

We hope that the above changes in response to your suggestions have addressed your concerns.

Kind regards,

Matthew Wells & Huzaifa Adamali